# *Treponema denticola* as a prognostic biomarker for periodontitis in dogs

**Daehyun Kwon**[1,2☯], **Kisuk Bae**[3☯], **HyeonJo Kim**[1], **Sang-Hyun Kim**[1], **Dongbin Lee**[1], **Jae-Hoon Lee**[1] *

**1** Institute of Animal Medicine, College of Veterinary Medicine, Gyeongsang National University, Jinju-si, South Korea, **2** May Veterinary Dental Hospital, Hannam-Dong Yongsan-Gu Seoul, Republic of Korea, **3** Bioscience Research Institute of BIOnME, Doyak-ro, Bucheon-si, Gyeonggi-do, Republic of Korea

☯ These authors contributed equally to this work.
* jh1000@gnu.ac.kr

**Data Availability Statement:** All relevant data are within the paper and its Supporting Information files.

## Abstract

Periodontal disease is one of the most common disorders in the oral cavity of dogs and humans. Periodontitis, the irreversible periodontal disease, arises progressively from gingivitis, the reversible inflammatory condition caused by dental plaque. Although the etiology of periodontitis has been widely studied in humans, it is still insufficient for the etiological studies on periodontitis in dogs. Many studies have reported that human periodontitis-related bacteria are putative pathogens responsible for periodontitis in dogs. However, most of these studies have focused on the appearance of a specific microbiome, and most of the cohort studies have insufficient sample sizes to generalize their results. In the present study, subgingival samples collected from 336 teeth were categorized into three groups at first, based on clinical outcomes (healthy, gingivitis, periodontitis). Subsequently, the periodontitis samples were further divided into three subgroups (early, moderate, and advanced periodontitis) according to the degree of periodontal attachment loss. Healthy and gingivitis were grouped as a reversible group, and the three subgroups were grouped as an irreversible group. To investigate trends of periodontopathic bacteria in the samples of dogs, a quantitative real-time polymerase chain reaction (PCR) was performed for quantification of 11 human periodontopathic bacteria as follows: *Aggregatibacter actinomycetemcomitans* (*Aa*), *Porphyromonas gingivalis* (*Pg*), *Tannerella forsythia*, *Treponema denticola* (*Td*), *Fusobacterium nucleatum*, *Prevotella nigrescens*, *Prevotella intermedia*, *Parvimonas micra*, *Eubacterium nodatum*, *Campylobacter rectus*, and *Eikenella corrodens*. The PCR results showed that *Aa* and *Pg*, the representative periodontopathic bacteria, were not significantly correlated or associated with the periodontitis cases in dogs. However, interestingly, *Td* was strongly associated with the irreversible periodontal disease in dogs, in that it was the most prevalent bacterium detected from the dog samples. These findings indicate that the presence and numbers of *Td* could be used as a prognostic biomarker in predicting the irreversible periodontal disease and the disease severity in dogs.

## Introduction

Periodontal disease is one of the most common disorders in the oral cavity of dogs and humans [1–5]. Periodontal disease is a group of illness occurring in the supportive tissues of

**Funding:** The author(s) received no specific funding for this work.

**Competing interests:** The authors have declared that no competing interests exist.

the teeth. There are two main types of periodontal disease, gingivitis and periodontitis, which are classified by disease severity and progression. Gingivitis, the initial stage of periodontal disease, is a reversible inflammatory condition that presents with swelling, redness, and bleeding from the gingiva. In contrast, periodontitis is a destructive and irreversible form of periodontal disease, resulting in tissue destruction in the periodontal ligament, cementum, alveolar bone, and gingiva [6]. Periodontitis caused by microbial biofilm often leads to host-mediated destruction of periodontal tissues, resulting in not only loss of teeth but also bloodstream bacterial infections [7]. It is therefore important to predict the prognosis of periodontal disease by identifying biomarkers that aid in the development of appropriate therapeutic interventions and subsequently evaluating the periodontal tissues after the treatment.

Although the etiology of human periodontitis has been widely studied for several decades, the etiology of periodontal disease in dogs has not been completely explained yet. However, Yamasaki et al [6] evaluated putative pathogens related to periodontal disease in dogs The subgingival microbiota implicated in periodontal disease in dogs is predominantly constituted by gram-negative anaerobes, and it shows significant similarities to those of humans [6,8,9]. Another study has indicated that periodontal diseases in dogs are more closely related to those in humans than to other animals [10].

Although recent studies have found that the oral microbiome in dogs is significantly different from the human oral microbiome [11,12], many previous studies have suspected that human periodontitis-related bacteria such as *Porphyromonas gingivalis (Pg)*, *Porphyromonas gulae*, *Prevotella intermedia (Pi)*, *Fusobacterium nucleatum (Fn)*, *Dialister pneumosintes*, *Actinobacillus actinomycetemcomitans*, *Campylobacter rectus (Cr)*, *Eikenella corrodens (Ec)*, *Tannerella forsythia (Tf)*, and *Treponema denticola (Td)* are putative periodontitis-related pathogens in dogs [13,14]. Most of these previous studies evaluated only the appearance of putative bacteria rather than trends in bacterial numbers and correlations of periodontitis stages in the dog. Furthermore, their limitations imposed by small cohort sizes have not been resolved [15,16]. It is necessary to analyze trends in bacterial numbers and associations between bacterial species and various periodontal conditions in dogs to overcome the limitations of previous studies.

Throughout the present study, we simply put all the clinical samples into two classification groups largely. Those were the reversible group (collectively, the healthy and gingivitis sample group) and the irreversible periodontitis group. These two (reversible and irreversible) groups were basically used for the comparison with each other to assess the tested 11 bacteria that were thought to be associated with periodontal disease. A quantitative real-time polymerase chain reaction (PCR) was performed for the quantification of these 11 periodontopathic bacterial species as follows: *Aggregatibacter actinomycetemcomitans* (*Aa*), *Pg*, *Tf*, *Td*, *Fn*, *Prevotella nigrescens* (*Pn*), *Pi*, *Parvimonas micra* (*Pm*), *Eubacterium nodatum* (*En*), *Cr*, and *Ec*. Based on results of PCR analysis, the correlation and association between the number of bacterial species and various periodontal conditions were analyzed. For the first time in veterinary field, the current study also investigated the association between bacterial combination to understand the interrelationships of reversible and irreversible periodontal conditions with different severities.

## Materials and methods

### Animals

Client-owned dogs examined for periodontal disease by MAY Veterinary Dental Hospital were used in the study. The present study was carried out in strict accordance with the recommendations in the Guide for the Care and Use of Laboratory Animals of the National Institutes of Health. The protocol was approved by the Institutional Animal Care and Use Committees (IACUC) of Gyeongsang National University (Approval no. GNU210813D0072). All surgery

was performed under sodium pentobarbital anesthesia, and all efforts were made to minimize suffering. The owners of the dogs were informed about the aim of the study and gave permission for sampling of their dogs. During the period January 2019 to February 2020, a total of 336 subgingival clinical samples were collected from 176 dogs. These dogs were aged over 1 year. Dogs were evaluated via a serum biochemical panel and CBC were performed on the day prior to scheduled periodontal treatment. They had not received any antibiotics over the past 3 months. Their body weights were less than 20 kg, a common finding in the veterinary field in South Korea.

## Sample collection

Clinical sample collections were performed under general anesthesia. The anesthesia protocol was briefly as follow: the premedication for the general anesthesia was glycopyrrolate (0.01 mg/kg, Mobinul; Myungmoon, Gyeonggi, Korea), subcutaneously, butorphanol (0.1 mg/Kg, Bu; Myungmoon, Gyeonggi, Korea) and midazolam (0.2 mg/Kg, Midacom; Myungmoon, Geyonggi, Korea), intravenously. Propofol (4 mg/Kg, Probio; Myungmoon, Geyonggi, Korea) was administered intravenously for the induction and the anesthesia was maintained with iso-flurane[f] at 1.5~2% and $O_2$ at 2 L/min followed by the placement of cuffed endotracheal tube. Lactated Ringer solution was administered through IV at a rate of 10 ml/Kg/hr through the procedure. Conductive-fabric patient warming system was placed under the dogs and they were monitored using combination monitoring equipment.

Sample collection sites were divided into rostral and caudal teeth of the oral cavity. Target teeth were mandibular and maxillary canine teeth known to be functionally important rostral teeth. The major masticatory teeth, maxillary fourth premolars teeth and mandibular first molars teeth, were also targeted as caudal teeth. Two samples were collected from each dog (one from the rostral target teeth and one from the caudal target teeth). One sample was collected if target teeth were missing in the oral cavity for some reasons.

All samples were obtained using a sterilized paper point International Organization for Standardization #30, which was gently inserted into the subgingival pocket at 6 points around each tooth for 30 seconds and transferred immediately into a sterilized transport tube (Fig 1A). Six paper points obtained subgingival plaque from six points around a tooth, and all six paper points were placed in a sterilized sample container, then a unique barcode number was assigned to the sample container and analyzed as one sample. Teeth were examined and evaluated by intraoral dental radiography and periodontal probing.

## Sample grouping by five stages of periodontal disease (PD)

Intraoral dental radiographs were obtained under general anesthesia and evaluated by the same veterinarian (DH-K) with a standard approach under the same conditions. According to the classification criteria of the *American Veterinary Dental College* [17], samples were divided into five PD groups based on clinical conditions, such as healthy (PD0), gingivitis (PD1), early periodontitis (PD2), moderate periodontitis (PD3), and advanced periodontitis (PD4). These five-stage PD grouping was based on the degree of attachment loss and gum condition. PD0 and PD1 included subjects without attachment loss. PD2, PD3, and PD4 included subjects with < 25%, 25%~50%, and > 50% attachment loss, respectively. A reversible group comprised PD0 and PD1, and an irreversible was group comprised PD2, PD3, and PD4 (Fig 1B).

## DNA extraction

DNA extraction was performed using an Exgene[TM] Cell SV kit (GeneAll Biotechnologies, Seoul, South Korea) according to the manufacturer's instructions. The paper point was treated

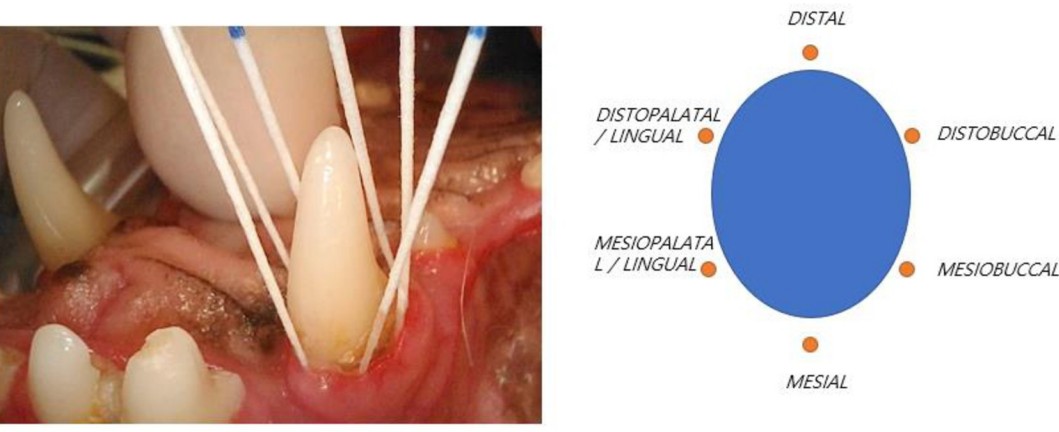

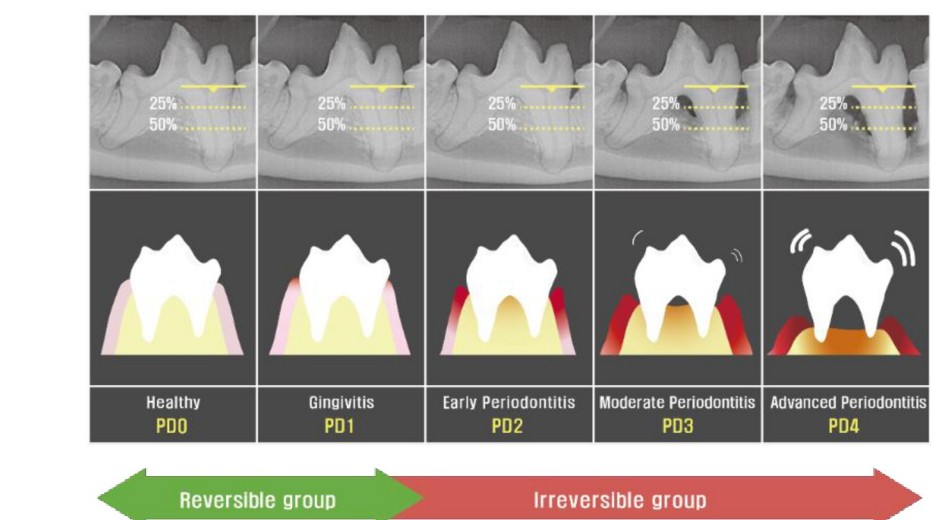

**Fig 1. Sample collection sites and grouping criteria.** (A) Sample collection sites are pointed at six positions around a tooth. (B) The subgingival plaque samples were classified into five periodontal disease (PD) groups based on the clinical conditions, such as healthy (PD0), gingivitis (PD1), early periodontitis (PD2), moderate periodontitis (PD3), and advanced periodontitis (PD4). The reversible group comprised PD0 and PD1, and the irreversible group comprised the other three PD groups (PD2, PD3, and PD4), which were classified by severity of periodontitis.

with 180 μL lysozyme at 30 mg/mL and incubated at 37˚C for 30 minutes. Proteinase K solution (20 μL of 20 mg/mL) and 200 μL of buffer BL were added to each sample followed by incubation at 56˚C for 30 minutes and 95˚C for 15 minutes. Subsequently, 200 μL of absolute ethanol was added, and the mixture was transferred into a column and centrifuged at 14,000 rpm for 1 minute. After washing the column with 600 μL of buffer BW, 700 μL of buffer TW was added. Next, 100 μL of buffer AE was used to elute DNA. DNA was quantified with a NanoDrop Spectrophotometer (Thermo Fisher Scientific, Inc., Waltham, MA, USA). DNA samples were stored at –20˚C before use.

## Quantitative real-time polymerase chain reaction assay

Targeted oral bacteria and primer/probe set sequence used for quantitative real-time PCR in this study are listed in Table 1. PCR amplification was performed in a reaction volume of 20 μL (Bioneer, Inc., Daejeon, South Korea). PCR cycling was performed using a CFX96™ Real-Time System (Bio-Rad Laboratories Inc., Hercules, CA, USA). Cycling conditions consisted of an initial denaturation step at 95˚C for 5 minutes, followed by 40 cycles of denaturation at 95˚C for 30 seconds, primer annealing at 60˚C for 40 seconds, and primer extension at 72˚C for 30 seconds. After completion of the cycling steps, a final extension step at 72˚C for 5

**Table 1. Targeted oral bacteria and primer/probe set sequence used for quantification real-time PCR.**

| Species and primer/probe | Primer/probe set sequence (5′ to 3′) | Length (base) | Amplicon size (bp) | Reference |
|---|---|---|---|---|
| *Aa* | | | 139 | This |
| AaLtF14 | CGGTGGAGAAGGAAATGATATTTATG | 26 | | study |
| AaLtR11 | ATTGCCGTTACGCTCAAATG | 20 | | |
| AaLtP13 | FAM-CCACACTATTACGGAACATAGCGGTG-BHQ-1 | 28 | | |
| *Pg* | | | 80 | This |
| PghaF14 | GCAGGGTCAGAAAGTAACGCTC | 22 | | study |
| PghaR13 | CGATCCGTTTTACTTCACGG | 20 | | |
| PghaP11B | HEX-CCGAGCGCAAAGAAGGCAGAA-BHQ-1 | 21 | | |
| *Tf* | | | 68 | This |
| TfKpF13 | CCGGCGGTTTCCTGTAGTAGA | 21 | | study |
| TfKpR12 | ACTTCGTCCGTTGCAGGGTT | 20 | | |
| TfKpP11 | TEXAS RED-CTCCCTTCACCCTCTCGCCG-BHQ-2 | 20 | | |
| *Td* | | | 98 | This |
| TdopF13 | CATCTCTTGATGCAGCCGAAG | 21 | | study |
| TdopR13 | GTCAGGGCTTACAACATAGTCGTC | 24 | | |
| TdopP01 | Cy5-TGGCGGAAGGAAAACAAGCC-BHQ-2 | 20 | | |
| *Fn.* | | | 73 | This |
| FnChF15 | GACATCTTAGGAATGAGACAGAGATG | 26 | | study |
| FnChR13 | CAGCCATGCACCACCTGTCT | 20 | | |
| FnChP12 | TEXAS RED-CAGTGTCCCTTCGGGGAAACCT-BHQ-2 | 22 | | |
| *Pn* | | | 79 | This |
| PngyF12 | GCAAGAACGTGATGACGGGA | 20 | | study |
| PngyR13 | ATTTCGCAGTCTTTGGGATCT TT | 23 | | |
| PngyP11 | Cy5-TTGCCAGGAAAACTTGCCGA-BHQ-2 | 20 | | |
| *Pi* | | | 103 | This |
| PipiF12 | CCACCAACGACAACCTTCCA | 20 | | study |
| PipiR13 | TCTACTGCTTCGAGCGCAC | 19 | | |
| Pi194P13H | HEX-CAAGACAATCTCCGACGGAACGTT-BHQ-1 | 24 | | |
| *Pm* | | | 201 | Nonnenmacher et al. [18] |
| PmF-30 | AAACGACGATTAATACCACATGAGAC | 26 | | |
| PmR-30 | ACTGCTGCCTCCCGTAGGA | 19 | | |
| Pm16S30 | TEXAS RED-TCAAAGATTTATCGGTGTAAGAAGGGCTCGC-BHQ-2 | 31 | | |
| *En* | | | 157 | This |
| EnglF01 | ATCCACAACAAAAGCGGCCT | 20 | | study |
| EnglR01 | AGGAATGTCCGGAGCAGGAA | 20 | | |
| EnglP01 | HEX-CAAACCAATCTGCAGCATGGG-BHQ-1 | 21 | | |
| *Cr* | | | 119 | This |
| CrgrF14 | GCGAAGTAGTGAGCGAAGAG | 20 | | study |
| CrgrR12 | GCCTGCGCCATTTACGATA | 19 | | |
| CrgrP01 | FAM-CAAGCGTGATCATCGACAAGGATAACA-BHQ-1 | 27 | | |
| *Ec* | | | 69 | Price et al. [19] |
| EcISRF-21 | AGGCGACGAAGGACGTGTAA | 20 | | Modified |
| EcISRR-21 | ATCACCGGATCAAAGCTCTATTG | 23 | | |
| EcISRP21 | Cy5-CGTGTAAGCCTGCGAAAAGCATCG-BHQ-2 | 24 | | |

**F**: Forward primer, **R**: Reverse primer, **P**: Probe.

**Table 2. Characteristic profiles of dogs that provided the subgingival plaque samples.**

| Characteristic | Reversible group | Irreversible group | $p$* |
|---|---|---|---|
| Age (years) | 9.36 ± 1.20 | 9.69 ± 1.40 | 0.113 |
| Male (%) | 54.7 | 58.3 | 0.585 |
| Weight (kg) | 6.44 ± 3.11 | 5.79 ± 3.04 | 0.061 |

Data of age and weight are given as mean ± standard deviation. Data of male are provided as percentage (%).

*Independent t-test.

minutes was performed. The normalized value of expression for each species was calculated as the ratio of the relative copy number of the reference species.

The qPCR performed with the primer-probe sets used in this study are as shown in the Table 1, and the data for assessment of specificity and sensitivity of the primer-probe sets detecting 11 bacteria from the subgingival plaques of dogs can be found in Supporting Information section.

## Statistical methods

All statistical analyses were performed using the MedCalc Statistical Software version 18.5 (MedCalc Software, Ostend, Belgium). The independence of characteristics of samples was analyzed by independent t-tests. The correlation between the number of bacteria and the severity of the periodontal condition was evaluated by Pearson's correlation coefficient. Associations of each bacterium or combination of bacteria between reversible and irreversible groups were analyzed using logistic regression. The significance level was set at $p < 0.05$.

## Results

### Characteristic profiles of dogs that donated the subgingival plaque samples

Characteristic profiles of dogs that donated the subgingival plaque samples are summarized in Table 2. The mean ages were 9.36 and 9.69 years for the reversible and irreversible groups, respectively. Males accounted for 54.7% and 58.3% in the reversible and irreversible groups, respectively. The mean body weights were 6.44 kg and 5.79 kg for the reversible and irreversible groups, respectively. An independent *t*-test was performed to show a statistical independence between these characteristics of the dogs and the reversible and irreversible groupings based on the dog's periodontal disease conditions. The statistical results showed that these dog's characteristics were not significantly associated with their periodontal disease conditions. Of the total 336 samples, 190 samples (PD0, 70; PD1 120 samples) were classified into the reversible group, and 146 samples (PD2, 77; PD3, 34; PD4, 35 samples) were classified as the irreversible group.

### Prevalence of the putative periodontopathic bacteria in dogs

**Overall bacterial prevalence.** The overall prevalence of the tested 11 bacteria detected in the reversible and irreversible groups is shown in Fig 2. *Fn*, *Pm*, and *Ec* showed a high prevalence over 50%, whereas *Aa*, *Pg*, and *Pn* showed a low prevalence less than 10%.

**Prevalence of the putative periodontopathic bacteria in the reversible and irreversible groups.** The prevalence of each bacterium was compared between the reversible and irreversible groups (Fig 3A). The prevalence of all bacterial species in the irreversible group was higher than that in the reversible group. Among the tested 11 bacterial species, *Aa*, *Tf*, *Td*, *Pn*, *Pi*, *Pm*, *En*, and *Cr* were detected two times more in the irreversible group compared with the

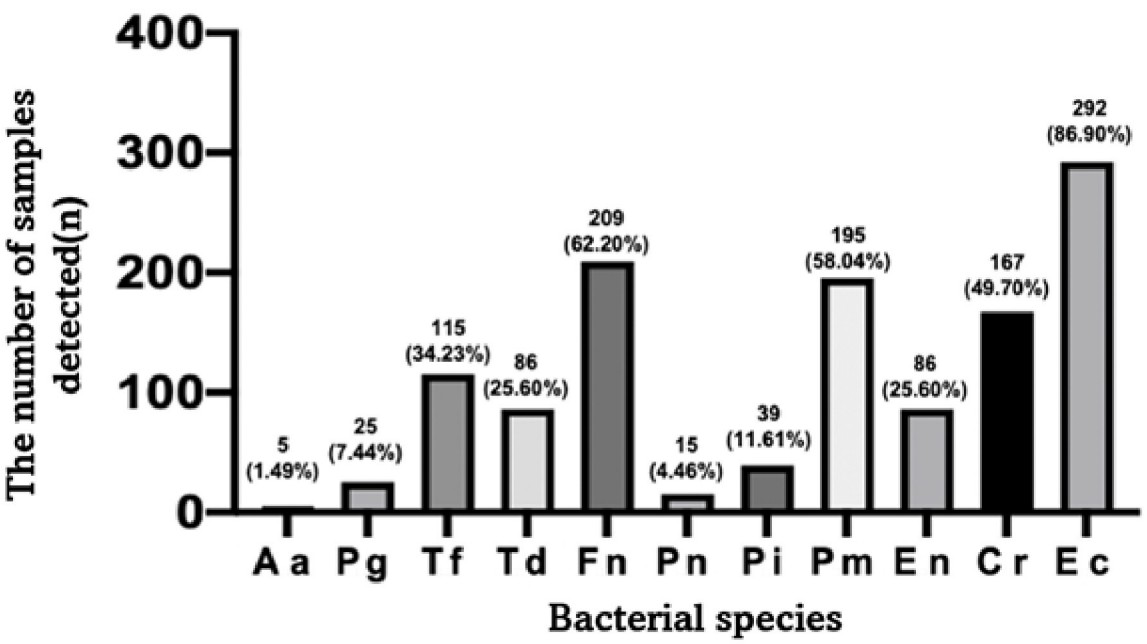

**Fig 2. Overall prevalence of bacteria in the reversible and irreversible groups.** Bars represent the number of samples that each bacterial species was detected and the percentage of the number of samples detected to the number of total samples (%). *Fn*, *Pm*, and *Ec* showed high prevalence at over 50%, whereas *Aa*, *Pg*, and *Pn* showed low prevalence at less than 10%.

reversible group. *Prevotella intermedia* showed the biggest difference in prevalence among the reversible and irreversible groups at 5.95 times. *Treponema denticola* showed the second biggest difference at 5.69 times.

**Prevalence of bacteria in the reversible and periodontal disease (PD) groups.** This study also compared the prevalence of each bacterium between the reversible group and each PD group (Fig 3B). Although the severity of periodontal disease was getting worse from PD2 to PD4, the prevalence of *Aa* and *Pg* showed a gradually decreasing trend. In contrast, the prevalence of *Fn*, *Pn*, *Pi*, and *Pm* continually increased along the increase of PD severity. Bacteria that showed increases over two times in their prevalence for all PD groups were *Tf*, *Td*, *Pn*, *Pi*, *En*, and *Cr*. were *Tf*, *Td*, *Pn*, *Pi*, *En*, and *Cr* showed more than double the difference in the prevalence of all PD groups. The fold change of *T. denticola*, the biggest change, was 7.68 times between the reversible and PD3 groups. *P. intermedia* showed the biggest difference between the reversible and PD4 groups at 9.31 times. The bacterium that showed the biggest increase in its prevalence between the reversible group and PD3 group (moderate periodontitis) was *T. denticola*. *P. intermedia* showed the biggest difference between the reversible group and PD4 group (advanced periodontitis).

## Mean number of bacteria

**Overall mean number of bacteria.** This study analyzed the mean number of all bacterial species in all groups (Fig 4). The mean number of *F. nucleatum* was 2.76E+07, which was the largest, whereas the mean number of *A. actinomycetemcomitans* was 6.21E+02, which was the smallest.

**Mean numbers of bacteria in the reversible and irreversible groups.** Mean numbers of bacteria in the reversible and irreversible groups were compared. Results are shown in Fig 5A.

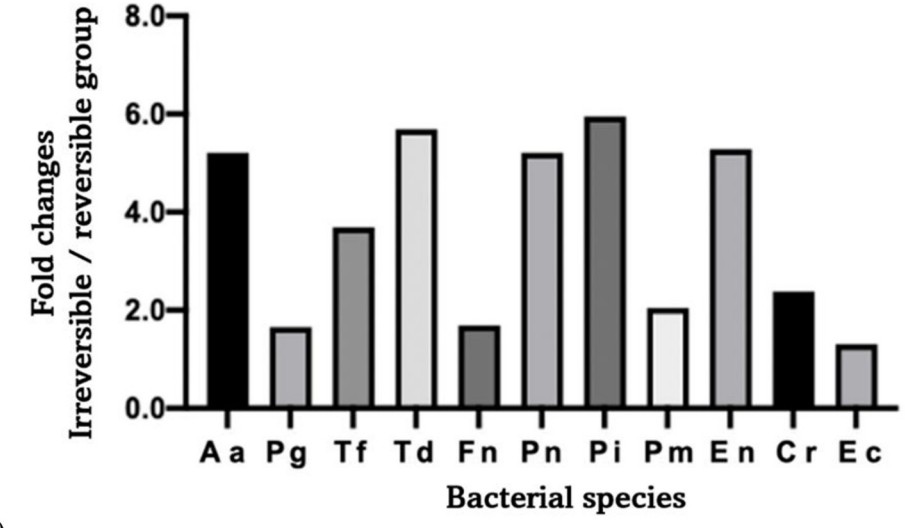

a)

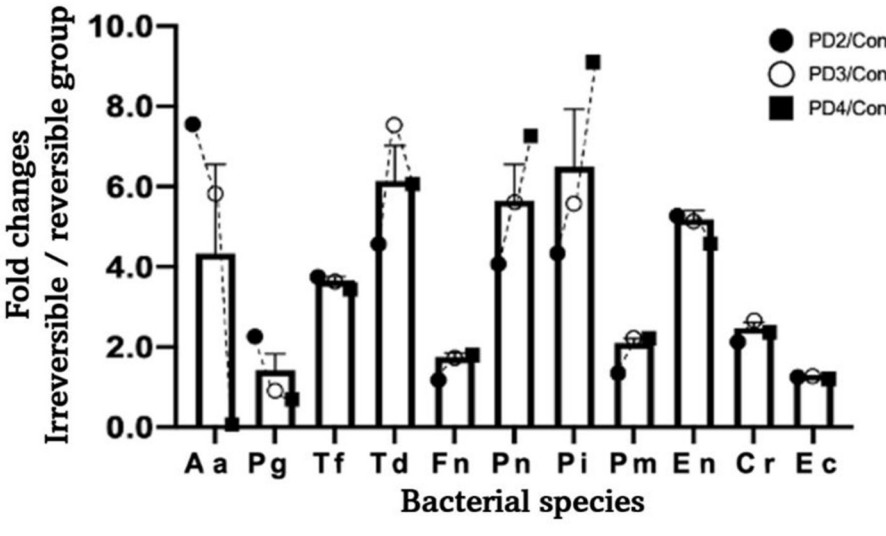

b)

**Fig 3. Prevalence of bacteria in comparison between groups.** (A) Comparison of fold changes between the reversible and irreversible groups.. (B) Comparison of fold changes between the reversible group and each periodontal disease (PD) group of the irreversible group. PD2: PD2 group, PD3: PD3 group, PD4: PD4 group, con: Reversible group.

Mean numbers of all bacterial species were increased in the irreversible group. Numbers of *Aa*, *Pg*, *Td*, *Pm*, *En*, and *Cr* in the irreversible group were higher over five times compared with those in the reversible group. However, *Aa* might not be associated with irreversible periodontal condition in dogs because of its extremely low overall bacterial number. The bacterium that showed the biggest difference in its number between the two groups was *T. denticola* (25.14 times). This result indicated a significant correlation between the number of bacteria and periodontitis, an irreversible condition.

**Mean numbers of bacteria in the reversible and PD groups.** Mean numbers of bacteria in the reversible group and each PD group were compared. Results are shown in Fig 5B. In the comparison between the reversible and PD2 groups, *Aa* and *Td* showed significant increases

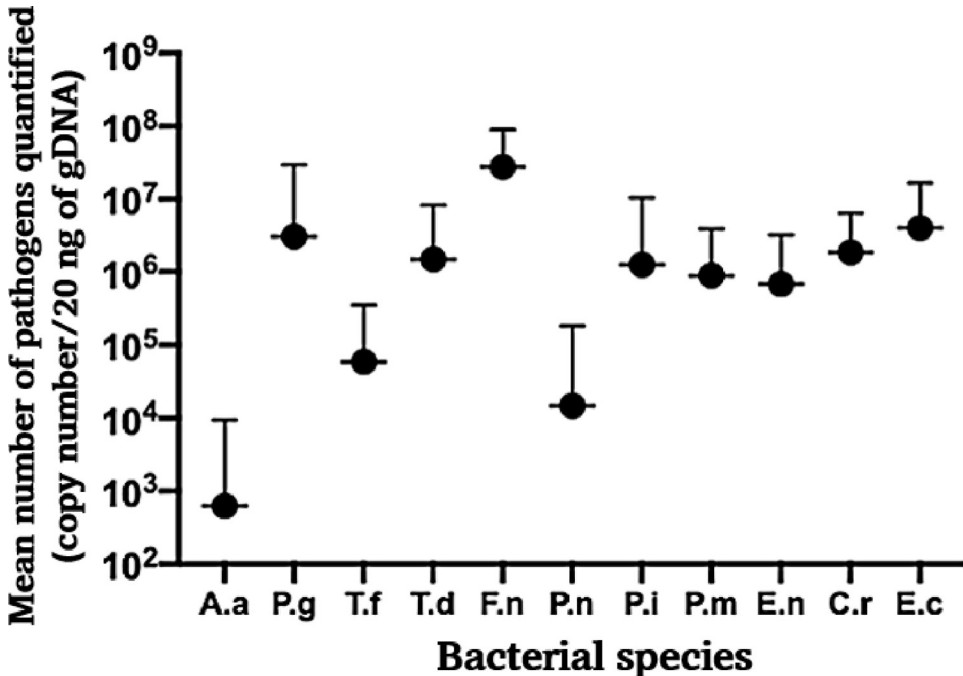

**Fig 4. Overall mean numbers of bacteria in all groups.** Bars show standard deviation. log = logarithm.

(24.31 times and 22.95 times, respectively). The difference of *Td*, 32.06 times, was unrivaled and overwhelming in comparison with the PD3 group. In the comparison of the reversible group and the PD4 group, *Td* and *En* showed the most overwhelming differences at 23.23 times and 21.62 times, respectively. *Aa*, *Pg*, and *Tf* showed significant differences in their numbers between the reversible and PD2 groups. However, the present study found that the number of bacteria continuously decreased as the severity progressed from PD3 to PD4. Specifically, *Aa* was not detected in the PD4 group. Conversely, *Pm*, *En*, and *Cr* increased continuously as severity progressed. Compared with the reversible group, the significant difference of *Td* in its number was observed in all PD groups. These results suggest that the number of bacteria except for a few could be correlated with the severity of periodontal condition, especially in the case of *T. denticola*.

### Correlation between the severity in the irreversible group and the number of bacterial species

Correlation analysis between the severity in the irreversible group and bacterial species was performed. All species, except *Aa* and *Pg*, showed reliable positive correlations with the irreversible group. In particular, *Td*, *Pm*, and *Cr* showed moderate correlations with Pearson's r > 0.4 (Table 3).

### Association between the number of each bacterial species and periodontal condition

Statistical methods were used to analyze associations between periodontal conditions and the number of bacteria using various comparisons (Table 4). Table 4 represents data that were p > 0.05. As shown in Table 4, the number of individual bacterial species was relatively increased, except for *Aa* and *Pg*, in the comparison between the reversible and irreversible

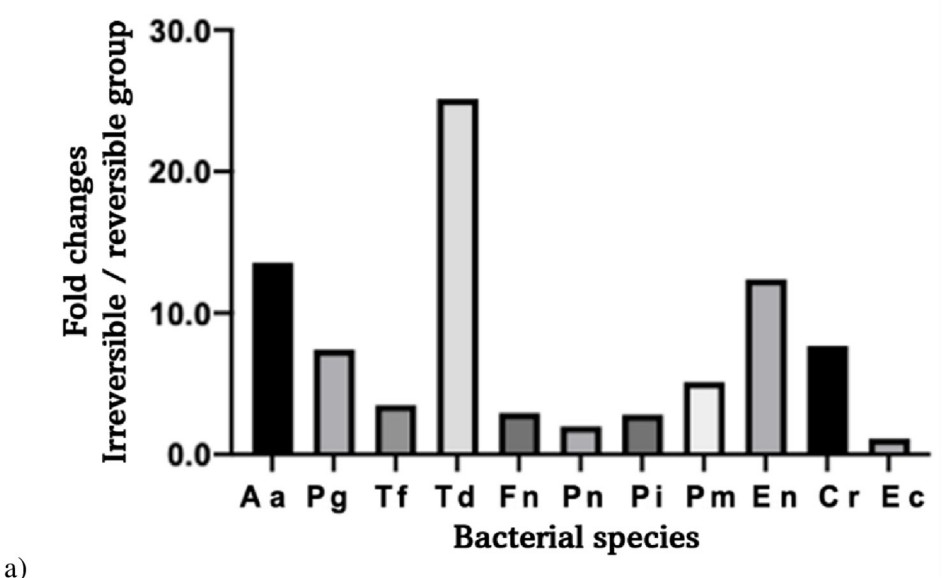

a)

b)

**Fig 5. Mean numbers of bacteria in comparison between groups.** (A) Comparison of fold changes between the reversible and irreversible groups. PD2: PD2 group, PD3: PD3 group, PD4: PD4 group, con: Reversible group.

groups. Similarly, *Aa* and *Pg* showed no significant reliability in associations between the reversible and each PD group. These facts indicated that the number of *Aa* and *Pg* had no association with periodontal conditions. Based on this, *A. actinomycetemcomitans* and *P. gingivalis* were excluded from further analysis.

## Association between the combinatory coexistence and numbers of bacterial species with the PD groups

With exception of *Aa* and *Pg*, as shown in Table 5, the combinatory coexistence and numbers of bacterial species among the remaining nine putative periodontopathic bacteria in the five-

**Table 3. Correlation between the severity of periodontal condition and the number of each bacterial species.**

| Species | Pearson's r | *p*-value |
|---------|-------------|-----------|
| *Aa* | 0.029 | 0.592 |
| *Pg* | 0.017 | 0.751 |
| *Tf* | 0.329 | <0.001 |
| *Td* | 0.430 | <0.001 |
| *Fn* | 0.357 | <0.001 |
| *Pn* | 0.144 | 0.008 |
| *Pi* | 0.310 | 0.010 |
| *Pm* | 0.428 | <0.001 |
| *En* | 0.354 | <0.001 |
| *Cr* | 0.413 | <0.001 |
| *Ec* | 0.260 | 0.006 |

Pearson's r: Pearson correlation coefficient.

Aa: Aggregatibacter actinomycetemcomitans, Pg: Porphyromonas gingivalis, Tf: Tannerella forsythia, Td: Treponema denticola, Fn: Fusobacterium nucleatum, Pn: Prevotella nigrescens, Pi: Prevotella intermedia, Pm: Parvimonas micra, En: Eubacterium nodatum, Cr: Campylobacter rectus, Ec: Eikenella corrodens.

stage PD groups showed significant associations with the irreversible group and with the respective PD groups. For the comparison between the reversible and irreversible groups, bacterial numbers of *Tf*, *Td*, *Pi*, *En*, and *Ec* showed significant associations ($p = 0.001$, $p < 0.001$, $p = 0.006$, $p = 0.009$, and $p = 0.004$, respectively). In the case of comparison between the reversible group and the PD2 group, numbers of *Tf*, *Td*, *En*, and *Ec* ($p < 0.001$, $p = 0.001$, $p = 0.002$, and $p = 0.002$, respectively) showed significant associations with the PD2 group. For the comparison between the reversible and PD3 groups, numbers of *Td*, *Fn*, *Pi*, *Pm*, and *Ec* also showed significant associations with the PD3 group ($p < 0.001$, $p = 0.019$, $p = 0.001$, $p = 0.020$, and $p = 0.043$, respectively). Numbers of *Td*, *Pi*, and *Pm* also showed significant associations with the PD4 group in the comparison between the reversible and PD4 groups ($p = 0.005$, $p < 0.001$, and $p = 0.014$, respectively). Numbers of *Td*, *Fn*, *Pi*, and *Pm* showed significant associations with PD3 + PD4 group in the comparison between the reversible and PD3 + PD4 groups ($p < 0.001$, $p = 0.005$, $p < 0.001$, and $p = 0.004$, respectively).

**Table 4. Relatively very low reliability of *Aa* and *Pg* in the Associations between the number of each bacterial species and the reversible and irreversible group and each PD group.**

| | Species | Odd ratio (95% CI) | *p*-value |
|---|---------|--------------------|-----------|
| Reversible group vs. irreversible group | *Aa* | 1.184 (0.942–1.488) | 0.148 |
| | *Pg* | 1.027 (0.969–1.088) | 0.182 |
| Reversible group vs. PD2 group | *Aa* | 1.227 (0.968–1.555) | 0.091 |
| | *Pg* | 1.045 (0.978–1.116) | 0.196 |
| Reversible group vs. PD3 group | *Aa* | 1.193 (0.890–1.600) | 0.238 |
| | *Pg* | 0.994 (0.891–1.108) | 0.907 |
| Reversible group vs. PD4 group | *Aa* | | N.A |
| | *Pg* | 1.012 (0.919–1.113) | 0.815 |
| Reversible group vs. PD3+4 group | *Aa* | 1.106 (0.826–1.482) | 0.499 |
| | *Pg* | 1.003 (0.929–1.083) | 0.935 |

Logistic regression.

**Table 5. Association between the combinatory coexistence and numbers of bacterial species with the periodontal disease groups.**

|  | Species | Odd ratio (95% CI) | *p*-value |
|---|---|---|---|
| Reversible group vs. irreversible group | Tf | 1.109 (1.042–1.182) | 0.001 |
|  | Td | 1.159 (1.086–1.237) | <0.001 |
|  | Pi | 1.105 (1.029–1.187) | 0.006 |
|  | En | 1.081 (1.019–1.146) | 0.009 |
|  | Ec | 1.161 (1.048–1.285) | 0.004 |
| Reversible group vs. PD2 group | Tf | 1.138 (1.060–1.221) | <0.001 |
|  | Td | 1.134 (1.050–1.224) | 0.001 |
|  | En | 1.114 (1.042–1.190) | 0.002 |
|  | Ec | 1.222 (1.078–1.386) | 0.002 |
| reversible group vs PD 3 group | Td | 1.288 (1.156–1.435) | <0.001 |
|  | Fn | 1.113 (1.018–1.352) | 0.019 |
|  | Pi | 1.207 (1.077–1.352) | 0.001 |
|  | Pm | 1.141 (1.021–1.276) | 0.020 |
|  | Ec | 1.283 (1.008–1.633) | 0.043 |
| Reversible group vs. PD4 group | Td | 1.152 (1.043–1.273) | 0.005 |
|  | Pi | 1.195 (1.082–1.319) | <0.001 |
|  | Pm | 1.145 (1.028–1.276) | 0.014 |
| Reversible group vs. PD3+4 group | Td | 1.212 (1.117–1.315) | <0.001 |
|  | Fn | 1.099 (1.029–1.175) | 0.005 |
|  | Pi | 1.193 (1.090–1.307) | <0.001 |
|  | Pm | 1.127 (1.039–1.223) | 0.004 |
| PD2 group vs. PD3 group | Td | 1.128 (1.039–1.226) | 0.004 |
| PD3 group vs. PD4 group |  |  |  |
| PD2 group vs. PD4 group | Pi | 1.097 (1.010–1.192) | 0.028 |

Logistic regression.

*P. intermedia* showed a high significance in the comparison between the reversible and PD3 groups, between the reversible and PD4 groups, and between the reversible and PD3 + PD4 groups. However, *T. forsythia* showed a significance in the comparison between the reversible and PD2 groups only. Based on the above results, *T. denticola* seemed to be a superior species to others in all different comparisons analyzed statistically in this study.

This study also statistically analyzed the association between each PD group. Only *T. denticola* was significantly associated with the PD3 group in the comparison between the PD2 and PD3 groups. *P. intermedia* was only significantly associated with the PD4 group in the comparison between the PD2 and PD4 groups. However, no bacteria showed significance in the comparison between the PD3 and PD4 groups.

## Discussion

Periodontitis is the most prevalent oral disease in humans. It also appears in approximately 44–63.6% of dogs [2]. The severity and prevalence of periodontitis in dogs increase with age [3]. This inflammatory oral disease is observed in approximately 80% of dogs aged 4 years [4], 82% of dogs aged 6–8 years, and 96% of dogs aged 12–14 years [5].

Periodontitis is an irreversible inflammatory periodontal condition that is always accompanied by anatomical destruction in the periodontium. It is classified into three grades as early, moderate, and advanced periodontitis according to the severity of attachment loss [17]. Since

the irreversibly destroyed periodontium cannot be reestablished, understanding the etiology of periodontitis is significantly important to predict and prevent the expected irreversible condition. Therefore, this study analyzed the trends of 11 bacterial species in dogs with reversible or irreversible periodontal conditions.

Dewhirst et al. [20] have obtained 353 canine oral bacterial taxa from 51 dogs and analyzed them. They found that only 16.4% of oral bacterial taxa were shared between dogs and humans and that 83.6% of identified taxa in dogs were novel. The 11 bacterial species focused in this study were included in 16.4% of oral bacterial taxa sharing with humans in the study. Other studies have also confirmed that canine microbiome is significantly different from human microbiome, revealing that bacteria not found in human subgingival plaque or did not have a high rate of prevalence were detected in dog subgingival plaque with significantly high percentages [11,21]. Wallis et al. [12] have longitudinally researched changes that occur in subgingival bacterial communities during the transition from health through gingivitis to the early stage of periodontitis in dogs. Their study suggested that a reduction in the proportion of health-associated species was a more evident indicator than the appearance of disease-associated species. Although the microbiome and environmental factors of dogs are relatively different from those of humans, most human periodontitis-related bacterial species are suspected as putative periodontitis-related pathogens in dogs [13,14].

Kumar et al. [22] have reported that higher number of species lost or gained is closely related to changed clinical status of the periodontal condition. They also suggested that the appearance of previously undetected bacterial species was not related to periodontitis. Rather, it was the result of microbial succession through fluctuation of newly abundant and/or meager bacterial species in subgingival space [23]. According to Griffen's study, it is significantly compelling to understand changed patterns of the number of putative periodontal pathogenic bacterial species and their associations through the periodontitis process in dogs. However, only a few studies have assessed periodontal pathogenic bacterial species and their associations in dogs. The majority of previous studies have analyzed only the difference in the prevalence of putative pathogens between limited severity of the disease such as health, gingivitis, and early periodontitis. Based on findings of the study by Griffen et al. [23], the prevalence of the 11 bacterial species in dogs, trends in their number changes, and their correlations and associations with irreversible periodontal condition with different severity were evaluated in this study for the first time in the veterinary field.

Nishiyama et al. [14] have assessed the presence of putative periodontal pathogens based on human periodontitis-related bacterial species in 40 dogs. It showed that *Pg*, *Cr*, *Aa*, *Pi*, *Tf*, *Fn*, and *Ec* were detected in 64%, 36%, 24%, 20%, 20%, 16%, and 12% of 25 dogs with periodontitis, respectively. *T. denticola* and *D. pneumosintes* were not detected in that study. Another study has analyzed 73 subgingival samples taken from dogs to assess the associations of red complex bacteria species in dogs [24]. It indicated that dogs with gingivitis or periodontitis were more likely to be infected with *T. forsythia* and *P. gingivalis*. Recently, Özavci et al. [25] have investigated the periodontal disease-related pathogens of dental plaque from 50 cats and 51 dogs with periodontal disease by PCR analysis. Their findings indicated that the periodontal disease-related pathogens, which showed high prevalence in the periodontal disease of cats and dogs, were *P. gingivalis* (cats 96%, dogs 88%), *P. nigrescens* (cats 90%, dogs 57%), and *P. gulae* (cats 70%, dogs 39%).

However, in this study, *A. actinomycetemcomitans* and *P. gingivalis* did not show any reliable correlations or associations with periodontitis. Numbers of all individual bacterial species, except *Aa* and *Pg*, in the irreversible group and each PD group were higher than those in the reversible group. Specifically, *Aa* showed extremely low overall prevalence and bacterial count. This study found that *Aa* and *Pg* did not have any reliable association or contribution to

periodontal disease in dogs. Except *Aa* and *Pg*, the other nine bacteria appeared to contribute to the periodontal process, although their mutual associations remained unclear.

Polkowska et al. [26] have analyzed gingival pocket microflora in 21 dogs experiencing moderate periodontitis (PD3) and advanced periodontitis (PD4). According to their study, the most commonly isolated pathogens in PD3 were *Streptococcus sanguis* (15.59%), *Peptostreptococcus* spp. (15.59%), *Escherichia coli* (12.84%), *Proteus mirabilis* (5.50%), *Veillonella* spp. (4.58%), and *Staphylococcus aureus* (4.58%). In PD4, *Peptostreptococcus* spp. (23.69%), *Streptococcus salivarius* (13.01%), *Veillonella* spp. (7.53%), *Actinomyces* spp. (6.16%), *Actinomyces viscosus* (4.10%), and *Staphylococcus aureus* (4.79%) were most often isolated [24]. The prevalence of *Peptostreptococcus* spp., which were reclassified as *Parvimonas* spp. in 2006, was significantly high in both PD3 and PD4 [26,27]. Similarly, the prevalence of *Pm* in PD3 and PD4 were over two times compared to that in the reversible group in our study. We also found that *Pm* was statistically associated with both PD3 and PD4 compared to the reversible group ($p < 0.05$).

The etiological findings from the periodontitis cases after analysis with the subgingival plaque samples in dogs were different from those that are widely studied in humans. Previous studies have described ecological succession and colonization of putative periodontopathic bacterial complexes in the subgingival plaque in a color-coded manner to demonstrate their abundance and the degree of association with periodontitis, including non-spore-forming anaerobes that usually form bacterial complexes in the subgingival plaque [28,29]. Subsequently, the keystone hypothesis has suggested that certain low-abundance pathogens could orchestrate inflammatory disease by remodeling a normally nonpathogenic microbiota into a dysbiotic one and altering host immune protective mechanisms [30,31]. According to these studies, *Pg* plays the strongest key role in the initiation and progression of human periodontitis, different from our findings. *Aa* is also strongly associated with human periodontitis by overwhelming the natural resistance of local host innate defense response and producing inflammatory cytokines that can result in connective tissue and bone loss [32].

One study has suggested that periodontal disease is not caused by a few isolated bacteria but by a community-wide effect of bacteria [12]. To confirm this, bacterial associations were analyzed using a combination of nine bacteria excluding *Ag* and *Pg* in this study. In the statistical evaluation of putative symbiotic associations of the combination of these nine bacteria between the reversible and irreversible groups, *Tf*, *Td*, *Pi*, *En*, and *Ec* showed significant associations with the irreversible group. Specifically, *T. denticola* showed a significantly strong reliability.

This study also evaluated the association of the combination of nine bacteria excluding *Aa* and *Pg* in the comparison between the reversible and PD groups. *Tf*, *Td*, *En*, and *Ec* were highly associated with the PD2 group, suggesting that the periosteum showed initial anatomical destruction. In the comparison between the reversible group and the PD3 group involving moderate periodontal destruction, *Td*, *Fn*, *Pi*, *Pm*, and *Ec* showed relatively high associations with the PD3 group. In the comparison between the reversible group and the PD4 group, the terminal stage of periodontitis, *Td*, *Pi*, and *Pm* were reliably associated with the PD4 group. *Td*, *Fn*, *Pi*, and *Pm* also showed strong associations with the PD3 + PD4 group in the comparison between the reversible group and the PD3 + PD4 group, suggesting a clinically more than moderate stage of periodontitis. *Tf* and *En* were related to the initial stage of irreversible periodontal condition. *Pi* and *Pm* were related to more than moderate stage of irreversible periodontal condition. However, *T. denticola* had a superior significance in contributing to periodontitis at different severity levels in dogs.

To understand whether any bacteria had strong associations with the progress of the irreversible periodontal condition, this study also analyzed the association between a combination

of nine bacteria and progression between the PD groups. *T. denticola* showed strong association with progression between PD2 and PD3. *P. intermedia* also showed a reliable association with progression between PD2 and PD4. However, no bacteria showed significant association with progression between PD3 and PD4. Therefore, *T. denticola* could be a predictable biomarker for the deterioration from early periodontitis to moderate periodontitis. In the aggravation from early periodontitis to advanced periodontitis, *P. intermedia* can be suggested as a predictable biomarker.

A previous study has investigated *Treponema* spp. in a gingival plaque from both healthy and periodontitis affecting 11 dogs to describe their occurrence and diversity. According to this phylogenetic analysis, *Td* and *Treponema maltophilum* (*Tm*) were mainly detected in the two groups in the diversity of treponema population. In addition, it indicated that *Td* and *Tm* are common in dogs regardless of periodontal status [33]. However, this finding is significantly different from that of the present study, which investigated 176 dogs. Although the overall prevalence of *Td* was 25.60%, *Td* showed the second biggest difference at 5.69 times in comparison of the prevalence between the reversible and irreversible group in this study. The mean number of *Td* in the irreversible group was also significantly higher than that in the reversible group, 25.14 times. Furthermore, this study reveals that *T. denticola* is most significantly correlated and associated with the irreversible group (periodontitis) in the above various statistical comparisons.

Several studies have revealed that *Porphyromonas gulae* is predominant in periodontal disease in dogs instead of *P. gingivalis*. Previous studies have noted that *P. gulae* is rarely found in humans but is commonly detected in animals with active periodontitis [7,34]. *P. gulae* is distinct from other related genera of *P. gingivalis*, which is known as a keystone pathogen in human periodontitis. Recently, it has been reported that *P. gulae* might be associated with periodontitis in dogs [35,36]. However, it is not clearly understood how *P. gulae* contributes to periodontal disease progression or how it cooperates with other bacterial species. Other studies have also shown that bacteria not found in human subgingival plaque or do not have a high rate of prevalence are significantly highly detected in subgingival plaque of dogs [11,12]. According to these studies, *Porphyromonas cangingivalis*, *Porphyromonas gingivicanis*, *P. gulae*, *Porphyromonas canoris*, *and Fusobacterium* sp. are highly prevalent in subgingival plaque of dogs, unlike in that of human beings. Although they might not contribute to the initiation or the progression of periodontitis in dogs because of their high prevalence through health to early periodontitis, we need to further evaluate the correlations and associations of these highly prevalent bacteria in dogs with periodontal disease, along with nine bacterial species analyzed in the present study.

Out of the putative 11 periodontopathic bacteria tested in this study, *A. actinomycetemcomitans* and *P. gingivalis* were not significantly associated in periodontal disease of dogs. However, the remaining nine others were significantly associated with periodontal diseases in dogs. Interestingly, among the nine PD-associated bacteria, *T. denticola* was the remarkable bacterium found in the irreversible group, which showed a correlation with the disease severity of periodontitis. Therefore, these findings suggest that *T. denticola* could be used as a prognostic biomarker in predicting the irreversible periodontal disease and the disease severity in dogs.

## Supporting information

**S1 Fig. Test results of qPCR sensitivity with the primer/probe set.**
(DOCX)

**S2 Fig. Standard curves generated from the *Ct* values for amplification of the target DNA of the red complex bacteria.**
(DOCX)

**S1 Table. Primer and probe set used for qPCR detection of 11 bacterial species from the subgingival plaques in the teeth of dogs.**
(DOCX)

**S2 Table. List of bacteria tested for the specificity of the qPCR with the primer/probe set.**
(DOCX)

**S3 Table. Test results of qPCR specificity with the primer/probe set.**
(DOCX)

## Acknowledgments

This report summarizes work contained within a thesis submitted by D.H. Kwon to fulfill the requirements for a MSc degree. The authors would like to thank Professor Frank J. M. Verstraete, Department of Surgical and Radiological Sciences, School of Veterinary Medicine, University of California, Davis for his helpful academic advice and encouragement.

## Author Contributions

**Conceptualization:** Daehyun Kwon, Kisuk Bae, Jae-Hoon Lee.

**Data curation:** Daehyun Kwon, Kisuk Bae, Jae-Hoon Lee.

**Formal analysis:** Kisuk Bae, Sang-Hyun Kim, Dongbin Lee, Jae-Hoon Lee.

**Methodology:** Kisuk Bae, HyeonJo Kim.

**Software:** Dongbin Lee.

**Supervision:** Sang-Hyun Kim, Dongbin Lee.

**Writing – original draft:** Daehyun Kwon.

**Writing – review & editing:** HyeonJo Kim, Jae-Hoon Lee.

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
