## [Decision Letter · Decision Letter 0]

25 Nov 2021

PONE-D-21-24139Treponema denticola as a prognostic biomarker for periodontitis in dogsPLOS ONE

Dear Dr. Lee,

Thank you for submitting your manuscript to PLOS ONE. After careful consideration, we feel that it has merit but does not fully meet PLOS ONE’s publication criteria as it currently stands. Therefore, we invite you to submit a revised version of the manuscript that addresses the points raised during the review process.

Please address the comments from all three reviewers, paying particular attention to R1 and R2's concerns about the data supporting the conclusions.

We look forward to receiving your revised manuscript.

Kind regards,

Catherine A. Brissette, Ph.D.

Academic Editor

PLOS ONE

Journal Requirements:

Reviewers' comments:

Reviewer's Responses to Questions

**Comments to the Author**

1. Is the manuscript technically sound, and do the data support the conclusions?

Reviewer #1: Partly

Reviewer #2: Partly

Reviewer #3: Yes

2. Has the statistical analysis been performed appropriately and rigorously? 

Reviewer #1: Yes

Reviewer #2: Yes

Reviewer #3: Yes

3. Have the authors made all data underlying the findings in their manuscript fully available?

Reviewer #1: Yes

Reviewer #2: Yes

Reviewer #3: Yes

4. Is the manuscript presented in an intelligible fashion and written in standard English?

Reviewer #1: Yes

Reviewer #2: No

Reviewer #3: Yes

5. Review Comments to the Author

Reviewer #1: This manuscript screened prognosis marker bacteria for dog periodontitis from dental plaque using quantification real-time PCR. The investigation of the dog periodontitis is a little and more investigation is required. This manuscript focused on the human periodontal pathogen in dog periodontitis. The viewpoint is interesting. Author had better to revise several points to improve manuscript.

1. Specificity of the primer/probe set of quantification real-time PCR was not clear.

The composition of the dental plaque bacteria in dog has not completely investigated. It is possible that the primers/probes cross reacted to closely related species to human periodontal pathogen. For example, the sequence of P. gingivalis and P. gluae shows very high similarity. In the present condition, the specificity of the detection of the species by the authors was not clear. Authors should describe the specificity of the primers used in this study.

2. Author showed the number of each organism in this paper. In the present condition, the percentage of the total microorganisms is unclear. It is helpful to understand the total number of the bacteria in the sample.

3. In the prevalence of the pathogen in the present study is significantly difference especially in P. gingivalis and A. actinomycetemcomitans. Authors mentioned about the condition of the disease such as PD3, PD4 for the comparison. The microflora changed according to the age and type of dog. Authors had better to include the condition of the dog for comparison of the prevalence.

Minor comments

1. Page 4, line 9. … alveolar bone, and gingiva. The description of periodontitis need the reference.

2. Page 4, line 16-17. However, many studies …disease in dogs [7]. Author only 1 report for the “many studies”.

Reviewer #2: This manuscript is straightforward and presents data demonstrating a strong association of Treponema denticola to be associated with canine periodontitis. A strength of this manuscript compared to previous literature is the larger sample size and collection of clinical samples from multiple sites around the tooth. Association analysis seems appropriate but the authors need to make sure to not overstate the relevance of this data.

1. The manuscript is fairly well written, but there are still quite a few minor grammatical edits required.

Also the authors should refer to " this current study" in places throughout the text of the discussion, as it is not always clear that they are referring to their own data presented in this manuscript.

2. Please clarify method for sample collection and DNA isolation. Were 6 individual paper points used as shown inFigure 1? Were these 6 paper points then pooled together to give 1 sample which was then treated with 180ul lysozyme as indicated the methods?

3. Most figures are depicting fold change while Figure 4 indicates absolute numbers of bacteria. indicate in the methods how these values were calculated ( compared to standard curve of each bacteria known numbers?)

If bacterial DNA was directly used ( or bacteria cultured) this information needs to be included.

4. table 4 refers to "each bacterial species" in the title, but only association for Aa and Pg is shown

5. THE methods do refer to the number of total dogs studied yet There is no mention of the number of animals that were divided into each classification group ( ie. PD1, Pd2 etc). How many animals were in the PD and P1 group ( reversible) or the PD2. Pd3 and Pd4 group ( irreversible). Were these classifications of the clinical sample collected made on each tooth collected or the overall radiographic appearance of the animal?

Reviewer #3: The manuscript presents an original and interesting point of view of the study of microbiota in canine periodontitis. My overall impression is that the section results is too long, and some parts should be deleted, considering that same data are presented also in the tables and figures.

Specific comments:

Table 1 could be moved to supplementary material.

Table 2, title ( and also headline in the results): “donated” is not appropriate, please substitute.

Figures: it is not necessary to comment the results. So delete sentences:

Figure 2, legenda: , starting from Fm, Pn, etc

Fig. 3: please delete last sentence, starting fromTreponema denticola

Fig 4: delete 3 sentences, starting from The mean number

Fig 5: delete starting from mean numbers

6. PLOS authors have the option to publish the peer review history of their article (what does this mean?). If published, this will include your full peer review and any attached files.

Reviewer #1: No

Reviewer #2: No

Reviewer #3: No

---

## [Author Response · Author response to Decision Letter 0]

5 Jan 2022

Catherine A. Brissette, Ph.D.

Dear Dr. Catherine A. Brissette

We thank the editor and referees of the “PLOS ONE” for taking their time to review our report. We have made some corrections and clarifications in the manuscript after going over the review’s comments. The changed are summarized below: 

Answer: It was approved by the Institutional Animal Care and Use Committees (IACUC) of Gyeongsang National University, an ethics committee related to animal experiments, and it was described in the animals section of materials and methods. 

Review Comments to the Author

Reviewer #1: 

This manuscript screened prognosis marker bacteria for dog periodontitis from dental plaque using quantification real-time PCR. The investigation of the dog periodontitis is a little and more investigation is required. This manuscript focused on the human periodontal pathogen in dog periodontitis. The viewpoint is interesting. Author had better to revise several points to improve manuscript.

1. Specificity of the primer/probe set of quantification real-time PCR was not clear.

The composition of the dental plaque bacteria in dog has not completely investigated. It is possible that the primers/probes cross reacted to closely related species to human periodontal pathogen. For example, the sequence of P. gingivalis and P. gluae shows very high similarity. In the present condition, the specificity of the detection of the species by the authors was not clear. Authors should describe the specificity of the primers used in this study.

Answer & Correction: We appreciate this point of specific comment asking us to provide the qPCR specificity/sensitivity data with the primer/probe set designed for this study. Accordingly, we provided additional data regarding the specificity/sensitivity test results in the Supporting Information section. 

In the Supporting Table S1 and S2, Porphyromoas gulae, the putative periodontopathic bacteria in dogs, was newly included to show the differential detection ability of the qPCR with the primer/probe set designed for specific detection of P. gingivalis (the human periodontopathic bacteria) in the subgingival plaque samples of dogs (Please see the highlighted portion of the tables).

In the revised manuscript, page 9, we added sentence “The qPCR performed with the primer-probe sets used in this study are as shown in the Table 1, and the data for assessment of specificity and sensitivity of the primer-probe sets detecting 11 bacteria from the subgingival plaques of dogs can be found in Supporting Information section.”

2. Author showed the number of each organism in this paper. In the present condition, the percentage of the total microorganisms is unclear. It is helpful to understand the 



Answer : In the present study, the total number of samples is 336, and the detection rate for each bacteria is expressed by dividing the number of samples detected for each bacteria. For example, in Figure 2, 5/336=1.49% of samples in which Aa: Aggregatibacter actinomycetemcomitans was detected among 336 samples were indicated.

3. In the prevalence of the pathogen in the present study is significantly difference especially in P. gingivalis and A. actinomycetemcomitans. Authors mentioned about the condition of the disease such as PD3, PD4 for the comparison. The microflora changed according to the age and type of dog. Authors had better to include the condition of the dog for comparison of the prevalence.

Answer: As mentioned in Table 2, the age, sex and weight of dogs under the reversible and Irreversible groups (PD3, PD4) were compared, but it was stated that there was no statistical difference. (12 pages)

Minor comments

1. Page 4, line 9. … alveolar bone, and gingiva. The description of periodontitis need the reference.

Answer & Correction: References in this content correspond to Reference No. 7 in the current manuscript. Accordingly, references were added to the manuscript and references 6 and 7 were changed. 

2. Page 4, line 16-17. However, many studies …disease in dogs [7]. Author only 1 report for the “many studies”.

According to reviewer 1's comment, the sentence has been modified as follows.

…..Yamasaki et al evaluated putative pathogens related to periodontal disease in dogs

 

Reviewer #2: 

This manuscript is straightforward and presents data demonstrating a strong association of Treponema denticola to be associated with canine periodontitis. A strength of this manuscript compared to previous literature is the larger sample size and collection of clinical samples from multiple sites around the tooth. Association analysis seems appropriate but the authors need to make sure to not overstate the relevance of this data.

1. The manuscript is fairly well written, but there are still quite a few minor grammatical edits required. Also the authors should refer to " this current study" in places throughout the text of the discussion, as it is not always clear that they are referring to their own data presented in this manuscript.

Answer and Corrections: Since the word "this study" may be confused with the current manuscript, our study was revised as "the current study" or "the present study" in the manuscript.

2. Please clarify method for sample collection and DNA isolation. Were 6 individual paper points used as shown in Figure 1? Were these 6 paper points then pooled together to give 1 sample which was then treated with 180ul lysozyme as indicated the methods?

Answer and correction: Yes, comments from Reviewer 2, the following sentence has been added to the sample collection section. The total number of samples is 336, and the detection rate for each bacteria is expressed by dividing the number of samples detected for each bacteria.

Page 4: Six paper points obtained subgingival plaque from six points around a tooth, and all six paper points were placed in a sterilized sample container, then a unique barcode number was assigned to the sample container and analyzed as one sample

3. Most figures are depicting fold change while Figure 4 indicates absolute numbers of bacteria. indicate in the methods how these values were calculated ( compared to standard curve of each bacteria known numbers?)

If bacterial DNA was directly used ( or bacteria cultured) this information needs to be included.

Answer: As mentioned in Table 2, the age, sex and weight of dogs under the reversible and Irreversible groups (PD3, PD4) were compared, but it was stated that there was no statistical difference. (12 pages)

4. table 4 refers to "each bacterial species" in the title, but only association for Aa and Pg is shown

Answers and corrections: Comments from Reviewer 2 Therefore, the contents of Table 4 have been corrected to the following sentences.

Table 4. Associations between the number of each bacterial species and the reversible and irreversible group and each PD group �Table 4. Relatively very low reliability of Aa and Pg in the Associations between the number of each bacterial species and the reversible and irreversible group and each PD group

Answers and corrections: In accordance with the amendments to Table 4, the relevant sentences in the document have been corrected to the following sentences.

Statistical methods were used to analyze associations between periodontal conditions and the number of bacteria using various comparisons (Table 4). � Statistical methods were used to analyze associations between periodontal conditions and the number of bacteria using various comparisons (Table 4)

5. THE methods do refer to the number of total dogs studied yet There is no mention of the number of animals that were divided into each classification group ( ie. PD1, Pd2 etc). How many animals were in the PD and P1 group ( reversible) or the PD2. Pd3 and Pd4 group ( irreversible). Were these classifications of the clinical sample collected made on each tooth collected or the overall radiographic appearance of the animal?



Answers and correction : The following was added to the results section. 

Page 13: …... Of the total 336 samples, 190 samples (PD0, 70; PD1 120 samples) were classified into the reversible group, and 146 samples (PD2, 77; PD3, 34; PD4, 35 samples) were classified as the irreversible group.

Reviewer #3: 

The manuscript presents an original and interesting point of view of the study of microbiota in canine periodontitis. My overall impression is that the section results is too long, and some parts should be deleted, considering that same data are presented also in the tables and figures.

Specific comments:

Table 1 could be moved to supplementary material.

Answer: If “Table 1” is not an issue for publication, we would like to leave it in the main manuscript. 

Table 2, title ( and also headline in the results): “donated” is not appropriate, please substitute.

Answers and corrections: As the reviewer's advice, the following corrections were made: “Table 2. Characteristic profiles of dogs that provided the subgingival plaque sample”

Figures: it is not necessary to comment the results. So delete sentences: Figure 2, legenda: , starting from Fn, Pn, etc

Answers and corrections: The part mentioned by the reviewer has been deleted.

Fig. 3: please delete last sentence, starting from Treponema denticola

Answers and corrections: The part mentioned by the reviewer was deleted and merged into a part of the related manuscript.

Fig 4: delete 3 sentences, starting from The mean number

Answers and corrections: The part mentioned by the reviewer has been deleted.

Fig 5: delete starting from mean numbers

Answers and corrections: The part mentioned by the reviewer has been deleted.

---

## [Editor Report · Decision Letter 1]

7 Jan 2022

Treponema denticola as a prognostic biomarker for periodontitis in dogs

PONE-D-21-24139R1

Dear Dr. Lee,

We’re pleased to inform you that your manuscript has been judged scientifically suitable for publication and will be formally accepted for publication once it meets all outstanding technical requirements.

Kind regards,

Catherine A. Brissette, Ph.D.

Academic Editor

PLOS ONE
---

## [Editor Report · Acceptance letter]

12 Jan 2022

PONE-D-21-24139R1 

*Treponema denticola* as a prognostic biomarker for periodontitis in dogs 

Dear Dr. Lee:

I'm pleased to inform you that your manuscript has been deemed suitable for publication in PLOS ONE. Congratulations! Your manuscript is now with our production department. 

Kind regards, 

on behalf of

Dr. Catherine A. Brissette 

Academic Editor

PLOS ONE